# Increased Secreted Frizzled-Related Protein 5 mRNA Expression in the Adipose Tissue of Women with Nonalcoholic Fatty Liver Disease Associated with Obesity

**DOI:** 10.3390/ijms23179871

**Published:** 2022-08-30

**Authors:** Laia Bertran, Marta Portillo-Carrasquer, Andrea Barrientos-Riosalido, Carmen Aguilar, David Riesco, Salomé Martínez, Amada Culebradas, Margarita Vives, Fàtima Sabench, Daniel Del Castillo, Cristóbal Richart, Teresa Auguet

**Affiliations:** 1Grup de Recerca GEMMAIR (AGAUR)—Medicina Aplicada (URV), Departament de Medicina i Cirurgia, Universitat Rovira i Virgili (URV), Institut d’Investigació Sanitària Pere Virgili (IISPV), 43007 Tarragona, Spain; 2Servei Medicina Interna, Hospital Universitari de Tarragona Joan XXIII, Mallafré Guasch, 4, 43007 Tarragona, Spain; 3Servei Anatomia Patològica, Hospital Universitari de Tarragona Joan XXIII, Mallafré Guasch, 4, 43007 Tarragona, Spain; 4Servei de Cirurgia, Hospital Sant Joan de Reus, Departament de Medicina i Cirurgia, Universitat Rovira i Virgili (URV), IISPV, Avinguda Doctor Josep Laporte, 2, 43204 Reus, Spain

**Keywords:** secreted frizzled-related protein 5, non-alcoholic fatty liver disease, obesity, adipose tissue-liver axis

## Abstract

Secreted frizzled-related protein 5 (SFRP5) is an anti-inflammatory adipocytokine secreted by adipocytes that seems to be linked with nonalcoholic fatty liver disease (NAFLD). We aimed to evaluate the role of the SFRP5-wingless-MMTV integration site family member 5a (WNT5A) pathway, closely related to adipogenesis, in subcutaneous (SAT) and visceral adipose tissues (VAT) and its relationship with obesity-related NAFLD. Our cohort was composed of 60 women with morbid obesity (MO), who underwent hypocaloric diet, subclassified according to their hepatic histopathology and 15 women with normal weight. We observed increased SFRP5 mRNA expression in VAT and lower WNT5A expression in SAT in MO compared to normal weight. We found elevated SFRP5 expression in nonalcoholic steatohepatitis (NASH) in SAT and in mild simple steatosis (SS) and NASH in VAT. We observed higher WNT5A expression in SS compared to normal liver in SAT, and a peak of WNT5A expression in mild SS. To conclude, we reported increased SFRP5 mRNA expression in SAT and VAT of NAFLD-related to obesity subjects, suggesting an implication of the SFRP5-WNT5A pathway in NAFLD pathogenesis, probably due to the adipose tissue-liver axis. Since the mechanisms by which this potential interaction takes place remain elusive, more research in this field is needed.

## 1. Introduction

Nonalcoholic fatty liver disease (NAFLD) is a complex metabolic disorder due to the interaction between genetic, hormonal and nutritional factors [1]. NAFLD is characterized by fatty accumulation in the liver in a higher proportion than 5% in the absence of secondary causes. This pathology concerns both hepatic simple steatosis (SS), the most benign state, and nonalcoholic steatohepatitis (NASH) [2,3], which is the severe form of the disease, since it can evolve to cirrhosis and hepatocellular carcinoma if left untreated [4]. NAFLD prevalence has increased in parallel with type 2 diabetes mellitus (T2DM) and obesity, two of its most important comorbidities [2,5,6].

Obesity is defined as an increased accumulation of fat, mainly in visceral adipose tissue (VAT) but also in subcutaneous adipose tissue (SAT), which can lead to chronic low-grade inflammation and oxidative stress [7,8,9]. The most important functions of adipose tissue are storing the body’s excess energy and acting as an endocrine organ for the regulation of metabolic homeostasis through the release of hormones and adipokines [10,11]. While VAT drains directly to the portal circulation of the liver [12] and seems to be directly associated with hepatic steatosis, liver inflammation and fibrosis [13,14,15], the relationship between SAT and NAFLD seems to lie in the transcriptome and macrophage phenotype [16].

Secreted frizzled-related protein 5 (SFRP5) is an anti-inflammatory cytokine produced and secreted by adipocytes of adipose tissue [17,18]. Considering the crosstalk of adipokines between adipose tissue and liver [19], some authors have linked SFRP5 functions to NAFLD [7,20]. This adipokine acts as an antagonist of the wingless-MMTV integration site (WNT) family member 5a (WNT5A), inhibiting WNT pathway activation. On the one hand, WNT signaling suppress adipogenesis during several stages of adipocyte differentiation [21]. On the other hand, in mature adipocytes, WNT pathway triggers proinflammatory, proliferative [20,22] and lipogenic outcomes [23]. These stimuli in adipose tissue result in the release of pro-inflammatory factors and free fatty acids that could contribute to the onset and progression of NAFLD, as described [24]. However, the molecular mechanism by which this possible interaction occurs is still unknown, and contradictory results have been reported regarding the function of the SFRP5 in adipose tissue [25,26]. The potential interrelationship between these pathways is graphically detailed in Figure 1.

Previous reports have demonstrated the protective anti-inflammatory role of SFRP5 in animal [17,25,26,35,36] and human models [26,37]. In contrast, WNT signaling is activated during liver fibrosis, although its mechanism is still unclear [38]. In this regard, further study to develop new anti-inflammatory strategies, such as SFRP5 enhancing action, may prevent NAFLD progression to NASH or even ameliorate NASH condition [39,40,41]. Moreover, since adipogenesis [42], insulin resistance [32] and oxidative stress have an important role in NAFLD development [43], proliferator-activated receptor gamma (PPARγ), a downstream mediator of the WNT cascade, is likely involved in the prevention of NASH progression [7,44].

Similarly, in our previous study, we evaluated SFRP5 serum levels and hepatic SFRP5, WNT5A and c-Jun N-terminal kinase (JNK), an intermediate of the noncanonical WNT5A pathway, mRNA expression in patients with morbid obesity (MO) and NAFLD. First, we reported higher serum levels of SFRP5 in a normal weight (NW) cohort than in MO patients and a higher hepatic expression of SFRP5 in patients with SS than in patients with normal liver (NL) or with NASH. Meanwhile, WNT5A and JNK hepatic expression increased in SS compared to NL, but higher levels were maintained in NASH [45]. In that study, we hypothesized that SFRP5 had a protective role in the first steps of NAFLD, trying to prevent its progression.

Based on the previous results and given that NAFLD development is susceptible to adipokines released from adipose tissue, the synthesis location of SFRP5, the aim of the present work was to evaluate the role of the SFRP5/WNT5A/PPARγ pathway in SAT and VAT and its link with NAFLD pathogenesis in a cohort of NW and MO women, with the latter subclassified according to their hepatic histopathology.

## 2. Results

### 2.1. Baseline Characteristics of the Patients

The anthropometric and biochemical characteristics of the studied cohort are shown in Table 1. Patients were first classified into NW (*n* = 15) and MO (*n* = 60) according to their body mass index (BMI), which were comparable in terms of systolic blood pressure (SBP), diastolic blood pressure (DBP) and low-density lipoprotein cholesterol (LDL-C). Next, those with MO were subclassified according to their hepatic histology as NL (*n* = 20), SS (*n* = 21) and NASH (*n* = 19) which were comparable in terms of weight, BMI, SBP, DBP, insulin, glycosylated hemoglobin (HbA1c), triglycerides (TG), cholesterol, high density lipoprotein cholesterol (HDL-C), LDL-C, alanine aminotransferase (ALT) and aspartate aminotransferase (AST).

Anthropometric measures and biochemical analyses showed that subjects with NW presented lower weight, BMI, HbA1c and TG than MO patients; furthermore, glucose, insulin, homeostatic model assessment method-insulin resistance (HOMA1-IR), AST and ALT were lower than in SS and NASH groups; also, gamma-glutamyltransferase (GGT) levels were reduced in comparison with NASH women. On the other hand, NW cohort had higher HDL-C compared to MO group and showed enhanced cholesterol levels than SS subjects. Regarding patients with MO, it has been shown that the NL cohort had lower levels of alkaline phosphatase (ALP) than SS patients, decreased levels of GGT compared to NASH group, and lower levels of glucose than SS and NASH patients. SS group had enhanced levels of ALP in comparison to NASH cohort.

### 2.2. Evaluation of Relative mRNA Abundance of SFRP5, WNT5A and PPARγ in SAT and VAT

First, we performed an assay to compare SFRP5, WNT5A and PPARγ mRNA expressions between SAT and VAT. As it can be observed in Figure 2, SFRP5 and WNT5A mRNA expressions in VAT were notably higher than in SAT, while there were no significant differences regarding PPARγ mRNA expression when both tissues were compared. It should be taken into account that SFRP5 mRNA expression was detectable in 85.5% of SAT and 97.2% of VAT samples, WNT5A was detected in 87.0% of SAT and 100% of VAT samples, and PPARγ was found in 100% of SAT and VAT samples.

### 2.3. Evaluation of Relative mRNA Abundance of SFRP5,WNT5A and PPARγ according to the BMI

In order to evaluate the role of SFRP5, WNT5A and PPARγ in SAT and VAT, we have studied the mRNA relative expression of these genes, first in a cohort classified according to the BMI into NW and MO.

The analysis of the SFRP5 mRNA according to the BMI presented non-significant differences between NW and MO groups in SAT (Figure 3A), however we found significantly increased expression of SFRP5 mRNA in MO subjects compared to NW (Figure 3B). Regarding WNT5A, we found reduced expression in patients with MO compared to NW group in SAT, as graphically represented in Figure 3C, but we did not observe significant differences between groups in VAT (Figure 3D). Concerning PPARγ mRNA expression, non-significant differences between analyzed groups were found (Figure 3E,F).

### 2.4. Evaluation of Relative mRNA Abundance of SFRP5, WNT5A and PPARγ according to NAFLD Presence

To ascertain if there were significant differences in SFRP5, WNT5A and PPARγ mRNA expression in adipose tissue (SAT and VAT) according to NAFLD presence, we first classified the MO cohort (the only one with liver biopsy for diagnosis) into NL and NAFLD. Regarding SFRP5, we reported a higher mRNA relative expression in NAFLD patients than in NL in SAT (Figure 4A) and also in VAT (Figure 4B). On the other hand, we did not observe differences between groups in SAT or in VAT concerning WNT5A (Figure 4C,D) and PPARγ (Figure 4E,F).

### 2.5. Evaluation of SFRP5, WNT5A and PPARγ Relative mRNA Abundance in Relation to NAFLD Grades

To achieve the main objective of the study, we evaluated SFRP5, WNT5A and PPARγ mRNA expression in adipose tissue (SAT and VAT) classifying the cohort into NAFLD grades. Focusing on SAT, our results showed an increase of SFRP5 mRNA expression in NASH group compared with NL (Figure 5A,B, this last one disaggregating the stage of SS into mild and moderate/severe (mod/sev)). In addition, we found that SFRP5 in SAT was enhanced in patients presenting ballooning (*p* = 0.046), one of the main NASH characteristics, compared to subjects without it. Regarding WNT5A expression in SAT, we found more elevated expression in SS compared to NL group (Figure 5C); whereas, when we separated the SS group into mild and mod/sev, no differences between groups were found (Figure 5D). Nevertheless, no differences were observed in PPARγ expression, as shown in Figure 5E,F.

When we analyzed the gene expression in VAT, we found higher SFRP5 mRNA expression in SS and NASH patients than NL (Figure 6A); but, when the SS group was disaggregated, the enhanced expression of SFRP5 was observed in mild SS stage and in NASH compared with NL group (Figure 6B). In reference to WNT5A mRNA expression, when we classified patients into NL, SS and NASH, there were no relevant differences (Figure 6C); however, when SS group was subdivided, we reported a significant enhanced expression of this gene in mild SS stage compared to the other groups, as shown in Figure 6D. In respect of PPARγ, we did not report meaningful differences, as graphically represented in Figure 6E,F.

Since we reported a peak of SFRP5 and WNT5A expressions in the SS mild stage in VAT samples, we decided to study the involvement of these genes focusing on the steatosis grades, regardless of inflammation or ballooning presence, in the analyzed cohort. Hence, we found significantly higher expression of SFRP5 in mild steatosis compared to absence (*p* = 0.001) or mod/sev degrees (*p* = 0.032); also, mod/sev group had an enhanced expression in comparison with the absence of steatosis cohort (*p* = 0.014). In terms of WNT5A expression, it was higher in mild than in mod/sev patients (*p* = 0.025).

### 2.6. Evaluation of Relative mRNA Abundance of SFRP5, WNT5A and PPARγ according to Metabolic Syndrome Parameters

Additionally, we wanted to evaluate SFRP5, WNT5A and PPARγ mRNA expression based on the presence of metabolic syndrome and its main components according to Alberti et al. classification [46]. Therefore, we found decreased expression of WNT5A in VAT (*p* = 0.034) and of PPARγ in SAT (*p* = 0.035) samples in patients presenting metabolic syndrome. In regard to T2DM, we only found lower levels of WNT5A mRNA abundance in SAT (*p* = 0.030) samples of the diabetic group. Moreover, we reported PPARγ decreased expression in SAT (*p* = 0.020) of the patients who presented with dyslipidemia. Nevertheless, we did not find significant differences in these genes regarding hypertension. In the case of SFRP5, we did not find significant associations with these disorders.

### 2.7. Correlations with SFRP5, WNT5A and PPARγ mRNA Expression in SAT and VAT of MO Cohort

To further investigate the possible relationships between NAFLD-related to obesity and SFRP5, WNT5A or PPARγ adipose tissue expression, associations have been analyzed with circulating cytokines, as shown in Table 2. In terms of SAT, interleukin (IL)-8, IL-17 and IL-22 correlated positively with SFRP5 mRNA expression, while this gene correlated negatively with resistin and adiponectin. On the other hand, in VAT, we reported a positive association of IL-1β, IL-13, IL-17 and plasminogen activator inhibitor-1 (PAI-1) with SFRP5 expression. Regarding WNT5A, in SAT, we could observe a positive correlation with IL-1β and adiponectin; while in VAT, a positive association was shown with IL-13, but monocyte chemoattractant protein-1 (MCP-1) correlated negatively. Concerning PPARγ we did not find significant associations.

## 3. Discussion

Recently, some authors have suggested that SFRP5 is a protective adipokine with an anti-inflammatory role in NAFLD pathogenesis [7,20]; however, inconclusive results have been reported. In this sense, given that the adipose tissue-liver axis seems to be relevant in the progression of NAFLD [16], in this study, we evaluated the SFRP5/WNT5A-PPARγ pathway in adipose tissue samples of women with different degrees of NAFLD associated to obesity.

First, we found a remarkably elevated mRNA expression of SFRP5 in VAT compared to SAT, which supports previous studies suggesting that SFRP5 is mainly synthetized and secreted by this fat depot [22,26]. Moreover, we also found increased WNT5A mRNA abundance in VAT than in SAT in the whole cohort, which is consistent with previous results [47,48]. In this sense, SFRP5 and WNT5A of VAT seem to be able to play relevant roles in the pathogenesis of NAFLD given the implication of this tissue in the adipokine and free fatty acid delivery into portal-vein circulation [12]. However, PPARγ presented nonsignificant different expression levels between tissues.

Next, we wanted to evaluate the mRNA abundance of SFRP5, WNT5A and PPARγ and their link with obesity, so we classified the cohort into NW subjects (control group) and MO patients. On the one hand, we reported enhanced expression of SFRP5 in the VAT of the MO group compared to the controls. Our results are consistent with previous reports, which suggested that SFRP5 expression is increased in adipose tissue of models with obesity, inducing adipogenesis [25,49,50,51]. However, we did not find a significant increased expression of PPARγ in the MO group, when this gene is the main regulator of adipogenesis [52]. On the contrary, our results differ from Ouchi et al., since they showed in animal models that SFRP5 expression in adipose tissue is reduced in response to severe obesity-related metabolic dysfunction [26]. These controversies could be explained because of the variability in the study model used or due to the very low-calorie diet that our subjects with MO followed three weeks prior to bariatric surgery, which was not followed by the NW group. This strict diet was indicated with the purpose of reducing the initial BMI before the surgery, and in this sense, Tan et al. and Schule et al. reported that children and adults with a caloric restrictive diet presented an increase in SFRP5 serum levels [40,53], proposing this adipokine as a possible biomarker for the anti-inflammatory effects of dietary interventions [40]. Additionally, and consistent with these facts, in an own previous study, we reported higher circulating levels of SFRP5 in patients with MO, who underwent the same hypocaloric diet, compared to NW subjects [45].

On the other hand, we only found a significant decrease in mRNA WNT5A expression in SAT of MO subjects compared to the NW group. Our results are difficult to explain, since no significant differences between subjects with obesity and without obesity have yet been reported in terms of WNT5A expression in SAT. In this sense, Catalán et al. found decreased, but nonsignificant, expression rates of WNT5A in SAT of subjects with obesity compared to lean subjects [28]. On the other hand, WNT5A pathway has been found to be remarkably more activated in VAT than in SAT in the presence of obesity [47,48], which is consistent with our results regarding WNT5A expression in VAT compared to that of the SAT in our whole cohort, the most part presenting MO. In any case, we need to mention again that our MO subjects underwent a very low-calory diet to induce weight loss before the surgery, and as Catalán et al. postulated, weight loss could lead to reduce circulating levels of the adipokine WNT5A [28]. Moreover, it is important to note that SAT and VAT depots presented different patterns of expansion during obesity [54], and given that the activation of WNT signaling by high-fat diet stimulates hypertrophy and overproliferation of adipocytes in SAT [55], we hypothesize that a very restrictive diet could induce the opposite effect inactivating WNT5A pathway.

Therefore, all these facts lead us to hypothesize that diet regulates adipokine levels, which are mostly secreted by adipose tissue, suggesting that dietary changes could have a direct impact on adipose tissue-gene expression and its endocrine function [56,57]. However, it needs to be further validated in other cohorts or other diets.

In the regard of PPARγ expression in adipose tissue, we did not find significant differences between MO and NW groups, similar to Torres et al. study [58]. The authors of this recent article made a systematic review of the prior studies reporting PPARγ expression in adipose tissue, and they found conflicting results: 12 previous studies have shown an increased mRNA expression of PPARγ in obesity, eight studies have reported a decreased expression, and four studies have found no differences compared to lean subjects. In this sense, although PPARγ is considered the “master regulator” of adipogenesis [52], the precise mechanism that PPARγ plays in the adipose tissue in human obesity remains unclear. The authors proposed variables that may cause discrepancy between these results, such as small sample size, variances in the reference gene used, different characteristics between the groups (gender, age range, insulin sensitivity, recent weight loss) and the type of the adipose tissue analyzed [58,59,60,61,62].

Subsequently, we aimed to investigate the involvement of SFRP5, WNT5A and PPARγ adipose tissue expression in NAFLD pathogenesis. To do so, we first classified women with MO into NL and NAFLD. In this sense, we observed increased expression of SFRP5 mRNA in NAFLD group compared to NL patients in both SAT and in VAT. Next, to better understand this finding, we divided NAFLD patients according to their hepatic histopathological degrees. First, we observed a higher expression of SFRP5 in NASH in SAT; and in VAT, we reported an increased expression of SFRP5 in mild SS and NASH stages. Regarding WNT5A, we found an elevated expression of WNT5A in SS in SAT, and a peak of expression of WNT5A in mild SS in VAT. To our knowledge, we are the first to investigate the SFRP5/WNT5A pathway in the adipose tissue regarding NAFLD presence in humans, and our results suggested that patients with NAFLD presented enhanced expression of SFRP5 in adipose tissues, specifically in the SAT of NASH patients and in the VAT of subjects with NASH or mild SS. Since SFRP5 has been reported to be an anti-inflammatory adipokine produced by adipose tissue [17,18,36,63], we hypothesize that the upregulation of SFRP5 mRNA abundance as NAFLD progresses may be due to an attempt to protect adipose tissue against inflammation and lipid accumulation and to counteract the inflammatory imbalance. This protective role of SFRP5 was already suggested in the liver in our previous study [45].

In other ways, the expression of SFRP5 in adipose tissue of NAFLD patients it could be also increased by the liver disease itself given the intercommunication between both tissues [16]. Since obesity has been suggested to induce SFRP5 expression in mice and humans [64], we hypothesize that NAFLD may result in the same effect by increasing metabolic imbalance. In this sense, Choudhary et al. reported a positive correlation between SAT volume and total adipose tissue volume with the degree of hepatic steatosis and severity, but none of the adipose tissue volumes correlated with NASH features such as lobular inflammation, ballooning, or fibrosis [65]. On the other hand, two previous reports also demonstrated increased VAT volume in NAFLD/NASH patients [66,67]. Nevertheless, we did not find significant differences in terms of PPARγ, the key regulator of adipogenesis [52], and we also found in SAT of subjects with mild SS, a higher expression of WNT5A, an inhibitor of pre-adipocyte differentiation [21]. Still, the increased expression of WNT5A in adipose tissue would make sense in firsts stages of steatosis, since in subjects with MO, there is an adipose tissue overproliferation mediated by WNT5A [54]. This excessive proliferation gives rise to a saturation of the adipose tissue depots that reduce insulin sensitivity [68,69]. In this regard, insulin resistance may impact the liver inducing hepatic steatosis [70]. In addition, our previous study reported an enhanced expression of WNT5A in liver samples of patients with NAFLD [41], giving consistency to a crosstalk between both tissues. The current findings and previous evidence suggest the possible upregulation of both competitor adipokines in the adipose tissue of patients with NAFLD, probably in the context of the metabolic dysfunction associated to obesity [30,43], highlighting the important role of adipose tissue-liver axis [71].

Next, we evaluated the implication on SFRP5, WNT5A and PPARγ in SAT and VAT according to metabolic syndrome presence and their components (T2DM, dyslipidaemia and hypertension). Hence, we found decreased expression of WNT5A in SAT in diabetic patients and reduced expression of WNT5A in VAT in metabolic syndrome patients. In this sense, although WNT5A has lipogenic effects, it has been reported that the inhibition of its pathway in the pancreas can lead to diabetes [69], which agrees with our results. Conversely, noncanonical WNT5A signaling has been found to play an essential role in obesity-induced VAT inflammation and metabolic dysfunction, which can promote insulin resistance under conditions of overfeeding [28,48,72]. However, this discrepancy may be mainly due to the hypocaloric diet followed by our patients before surgery and, as previously mentioned, this fact should be studied in subjects under other conditions.

Focusing on PPARγ, an insulin-sensitizing protein involved in the inhibition of lipogenesis [73,74], it makes sense that in our study, PPARγ levels were lower in SAT of metabolic syndrome and dyslipidaemic patients compared to that in healthy subjects. Unfortunately, in this analysis, we did not find significant variances in SFRP5 mRNA abundance in adipose tissues based on the studied comorbidities. In this case, significant differences in the analyzed transcripts were mostly present in SAT, which can be explained given that SAT protects other tissues from lipotoxicity, acting as a “buffer” of dietary lipid intake; also, a different profile of secreted adipokines has been reported between SAT and VAT [75,76].

Finally, we performed a correlation analysis between SFRP5, WNT5A and PPARγ in both tissues and some NAFLD-related cytokines. On the one hand, we observed a positive correlation between SFRP5 mRNA in SAT and proinflammatory cytokines, such as IL-8 and IL-17, which presented high levels in patients with fatty liver disease [77,78], consistent with our results. Additionally, it was positively correlated with IL-22, whose role in NAFLD is unclear, since some studies have indicated a possible protective action during liver injury [79,80], while others have reported that it can promote NAFLD development [81]. On the other hand, we reported a negative correlation between SFRP5 mRNA in SAT and resistin and adiponectin. Resistin is a circulating proinflammatory cytokine that is used to increase in obesity [82] and NAFLD [83], while adiponectin is an anti-inflammatory adipokine that tends to decrease in NAFLD patients compared to that in control subjects [84,85]. These correlations are difficult to explain and give rise to controversies, as we have found in this study that SFRP5, an anti-inflammatory cytokine, was increased in SAT samples from NASH patients. In addition, of note, this analysis was carried out in SAT, not in VAT, which is the major inductor of low-grade chronic inflammation in obesity [86]. Moreover, a differential pattern of functionality and adipokine production has been demonstrated between both tissues due to the clearance function of SAT [87,88].

On the subject of SFRP5 mRNA expression in VAT, we found a positive correlation with the proinflammatory cytokines IL-1β, IL-17 and PAI-1 levels, whose increase has been linked to obesity and NAFLD [78,89,90], and with IL-13, a cytokine with an anti-inflammatory role that used to be increased in obesity [91], which presents beneficial effects in hepatic diseases [92], as we previously hypothesized that SFRP5 does. All these results are consistent with what was obtained in this study, since SFRP5 mRNA expression in VAT is significantly elevated in MO compared with NW, and it was higher in NAFLD.

In relation to WNT5A mRNA in SAT, we observed a positive correlation with IL-1β, a proinflammatory cytokine related to NAFLD progression [89], and with the anti-inflammatory adiponectin [84,85]. This last association is difficult to explain and is the same that happened with SFRP5 in VAT, due to the peculiar peak of expression that these proteins present in the mild SS group. Focusing on WNT5A mRNA in VAT, we found a positive association with IL-13, which was mentioned above in the SAT analysis. Moreover, we found a negative association between WNT5A mRNA in VAT and MCP-1, a proinflammatory adipokine [93] that presents higher levels in accordance with an increase in BMI, leading to low-grade chronic inflammation [94]. This trend is similar to that of SFRP5 correlations with proinflammatory cytokines.

In summary, SFRP5 and WNT5A in adipose tissue could be attractive therapeutic targets to prevent lipogenesis, inflammation and the consequent liver damage [22], since it seems that their synthesis is enhanced when NAFLD begins in obese subjects [40,53]. However, more research in this field is needed to determine the specific molecular mechanism by which SFRP5 and WNT5A in adipose tissue are linked with the liver histology. In this study, our cohort of women made it possible to relate SFRP5, WNT5A and PPARγ mRNA abundance in SAT and VAT with obesity and NAFLD. Nevertheless, these results cannot be extrapolated to other obesity groups, patients who do not undergo a caloric restriction diet or men. In addition, it is important to mention that the subjects we studied were able to drink a maximum of 10 g of alcohol per day, so we cannot completely rule out any effect on the levels of the adipokines studied or even on liver histology. Additionally, we are not able to explain whether this association underlies a causal relationship or whether it is merely an epiphenomenon within the context of these complex pathogenic mechanisms. Therefore, further studies would validate our results.

## 4. Materials and Methods

### 4.1. Subjects

The institutional review board (Institut Investigació Sanitària Pere Virgili (IISPV) CEIm; 23c/2015) approved this research. The studied cohort is composed of 60 Caucasian women with MO (BMI > 40 kg/m^2^), and a control group of 15 Caucasian women with NW (BMI 19–25 kg/m^2^); all of whom gave written informed consent. Liver, SAT and VAT biopsies of MO subjects were obtained during planned bariatric surgery, while SAT and VAT biopsies of NW patients were obtained during other planned surgeries (such as discectomy or cholecystectomy). Hepatic biopsies were indicated only when a clinical diagnosis was needed. Patients who underwent bariatric surgery (MO subjects) followed a very low caloric diet for three weeks before the intervention, whereas NW cohort did not undergo any specific diet prior to the surgery.

The study exclusion criteria were women who: (1) had an intake of alcohol higher than 10 g/day or other toxins; (2) had acute or chronic hepatic illness of other etiologies, inflammatory, infectious or neoplasic diseases; (3) were menopausal or undergoing contraceptive treatment; and (4) were treated with antibiotics in the previous 4 weeks.

In this study, we include only women to evaluate a homogenous cohort to avoid the interference of confounding factors such as gender. It is well known that men and women differ substantially regarding body composition, energy imbalance and hormones. Moreover, several studies showed sex-specific differences in lipid and glucose metabolism [95,96] and adipokine profile [97].

### 4.2. Hepatic Histological Evaluation

Women with MO, from which a liver biopsy was obtained, were classified by one experienced pathologist according to hepatic histopathological classification following the Brunt criteria described elsewhere [98,99] into patients with NL histology (*n* = 20) or NAFLD (*n* = 40); furthermore, NAFLD cohort was subclassified as SS (micro/macrovesicular steatosis without inflammation or fibrosis, *n* = 21) or NASH (Brunt Grades 1–2, *n* = 19). It should be noted that none of the patients with NASH in our cohort presented fibrosis.

### 4.3. Sample Size

This work is mainly focused on defining the specific role of SFRP5 in NAFLD. In this sense, sample size was calculated using the ARCSINUS approach, with an α risk of 0.05 and a β risk of 0.1 in a bilateral contrast. Hence, a minimum of 15 subjects were needed in the first group (NW) and 45 in the second (MO) to detect differences between them with statistical significance.

### 4.4. Biochemical Analyses

All the participants underwent physical, anthropometric and biochemical assessments. Blood extractions were performed by specialized nurses through a BD Vacutainer^®^ system, after overnight fasting before the surgery. Venous blood samples were obtained using empty and ethylenediaminetetraacetic acid coated tubes, which were, respectively, separated into serum and plasma aliquots by centrifugation (3500 rpm, 4 °C, 15 min) (Fisher Scientific SL, Madrid, Spain). Biochemical parameters were analyzed using a conventional automated analyzer. Insulin resistance was estimated using HOMA1-IR.

IL-1β, IL-6, IL-7, IL-8, IL-10, IL-13, IL-17, IL-22, tumor necrosis factor α (TNF-α), PAI-1, MCP-1, adiponectin and resistin were determined using multiplex sandwich immunoassays and the MILLIPLEX MAP Human Adipokine Magnetic Bead Panel 1 (HADK1MAG-61K, Millipore, Billerica, MA, USA) and MILLIPLEX MAP Human High- Sensitivity T Cell Panel (HSTCMAG28SK, Millipore, Billerica, MA, USA), and the Bio- Plex 200 instrument (Bio-Rad Laboratories SA, Madrid, Spain). at the Center for Omic Sciences (Universitat Rovira i Virgili), according to the manufacturer’s instructions.

### 4.5. Gene Expression Analysis 

SAT, VAT and hepatic biopsies were collected in tubes with RNAlater (Qiagen, Hilden, Germany) during the surgery and were conserved at 4 °C and then, processed and stored at −80 °C. The RNeasy mini kit (Qiagen, Barcelona, Spain) was used to extract total RNA from SAT and VAT tissues. Reverse transcription to cDNA was performed with the High-Capacity RNA-to-cDNA Kit (Applied Biosystems, Madrid, Spain). Real-time quantitative PCR was assessed with a TaqMan Assay predesigned by Applied Biosystems for the detection of SFRP5 (Hs00169366_m1), WNT5A (Hs00998537_m1), PPARγ (Hs01115513_m1) in adipose tissue samples. The expression of each gene was first normalized with the expression of glyceraldehyde-3-phosphate dehydrogenase (GAPDH, Hs02786624_g1), and then standardized with the control group. All reactions were carried out in duplicate in 96-well plates using the QuantStudio™ 7 Pro Real-Time PCR System (Applied Biosystem, Madrid, Spain).

### 4.6. Statistical Analyses

The SPSS/PC+ for Windows statistical package (version 27.0; SPSS, Chicago, IL, USA) was used to analyze the data. The Kolmogorov-Smirnov test was performed to assess the distribution of variables. Variables were reported as the median and the 25–75th percentile or mean and standard error of the mean when we are analyzing relative expressions of genes. The different comparative analyses between groups were performed using the nonparametric Mann-Whitney U test. The strength of the association between variables was calculated using Spearman’s ρ correlation test. *p*-values < 0.05 were considered statistically significant. Graphics were performed using GraphPad Prism (version 7.04; San Diego, CA, USA).

## 5. Conclusions

In conclusion, we reported and elevated SFRP5 mRNA expression in adipose tissue of patients with NAFLD-related to obesity. This study suggested that the increased expression of SFRP5 in VAT seems to be induced by obesity, and by NAFLD pathogenesis in SAT and VAT, with a potential protective role against metabolic imbalance. On the contrary, we suggest that the increased WNT5A expression in SAT of SS subjects and in VAT of mild SS subjects may promote liver damage. Therefore, we hypothesize that SFRP5/WNT5A pathway in adipose tissue might play an important role in NAFLD-related to obesity pathogenesis through the adipose tissue-liver axis. However, further studies are needed in this field to establish the molecular mechanism and validate these hypotheses.

## Figures and Tables

**Figure 1 ijms-23-09871-f001:**
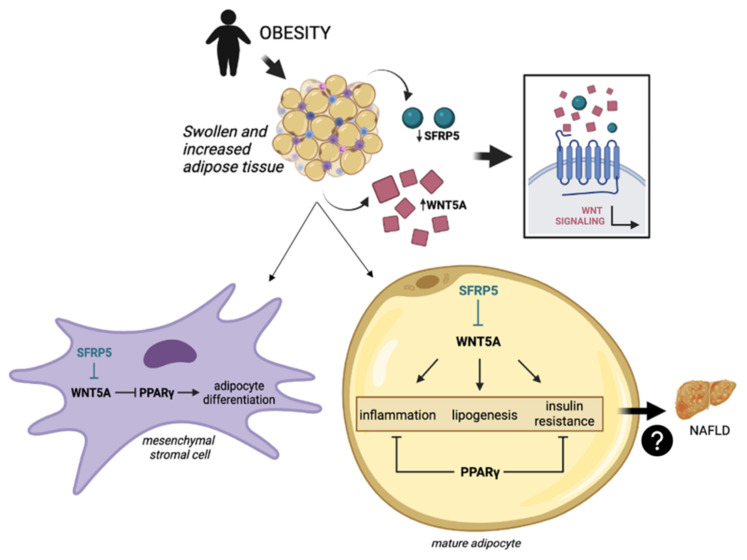
Obesity, the main comorbidity of NAFLD [2,5,6], causes chronic low-grade inflammation of the adipose tissue, leading to an upregulation of WNT5A [27,28] and a decrease in SFRP5 [26]. In mesenchymal stromal cells, WNT5A inhibits the adipogenesis by blocking PPARγ [21]. In mature adipocytes, WNT5A triggers an inflammatory response, lipogenic outcomes and insulin resistance [23,29,30], while PPARγ exhibits anti-inflammatory properties and acts as an insulin sensitizer [31,32]. These signs of metabolic breakdown could negatively affect the liver, making it susceptible to NAFLD occurrence [33,34]. NAFLD, nonalcoholic fatty liver disease; SFRP5, secreted frizzled-related protein 5; WNT, wingless-MMTV integration site; WNT5A, WNT family member 5a; PPARγ, peroxisome proliferator-activated receptor gamma.

**Figure 2 ijms-23-09871-f002:**
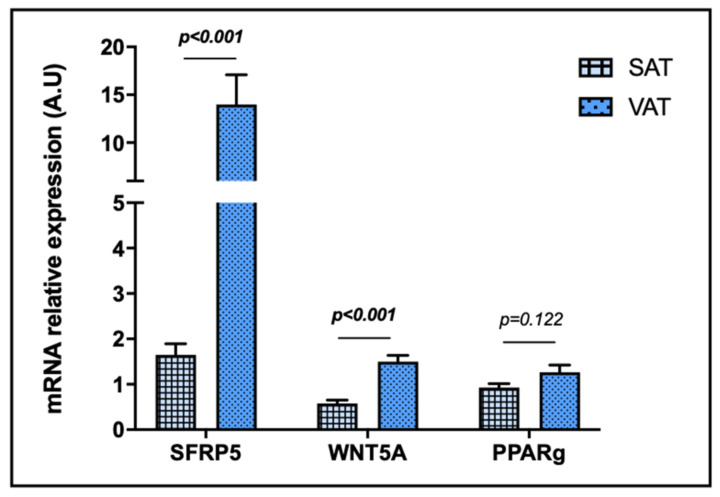
Differential relative mRNA abundance of SFRP5, WNT5A and PPARγ in subcutaneous and visceral adipose tissue of the entire unclassified cohort. The data were normalized by NW group and expressed as mean standard error of the mean (SEM). SAT, subcutaneous adipose tissue; VAT, visceral adipose tissue; SFRP5, secreted frizzled-related protein 5; WNT5A, WNT family member 5a; PPARγ, peroxisome proliferator-activated receptor gamma; A.U arbitrary units. *p* < 0.05 was considered statistically significant (bold).

**Figure 3 ijms-23-09871-f003:**
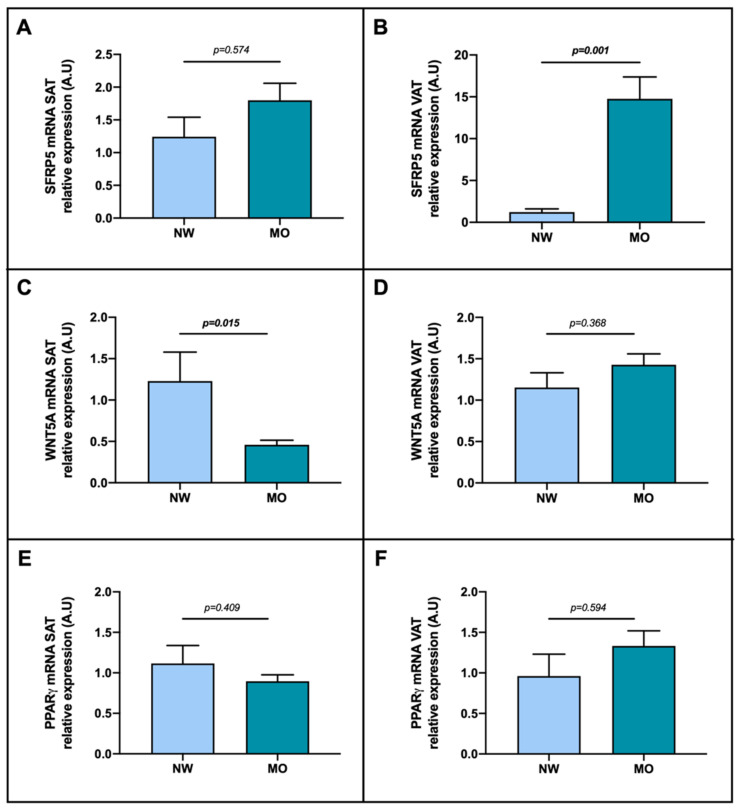
Differential relative mRNA abundance of (**A**) SFRP5, (**C**) WNT5A and (**E**) PPARγ in subcutaneous adipose tissue and (**B**) SFRP5, (**D**) WNT5A and (**F**) PPARγ in visceral adipose tissue of women classified depending on their BMI in NW and MO. The data were normalized by NW group and expressed as mean standard error of the mean (SEM). SAT, subcutaneous adipose tissue; VAT, visceral adipose tissue; SFRP5, secreted frizzled-related protein 5; WNT5A, WNT family member 5a; PPARγ, peroxisome proliferator-activated receptor gamma; NW, normal weight; MO, morbid obesity; A.U arbitrary units. *p* < 0.05 was considered statistically significant (bold).

**Figure 4 ijms-23-09871-f004:**
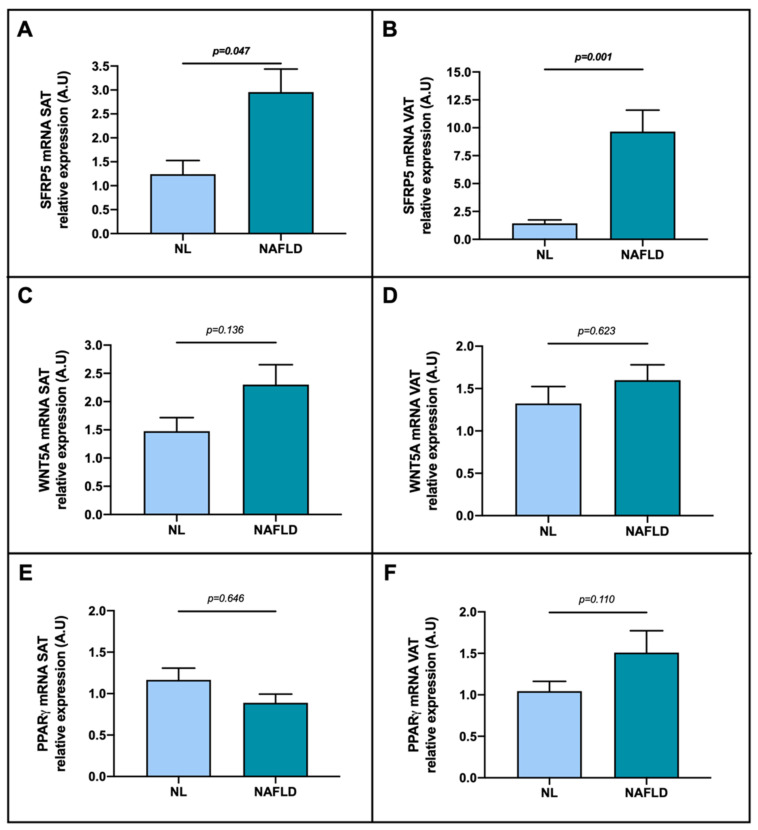
Differential relative mRNA abundance of (**A**) SFRP5, (**C**) WNT5A and (**E**) PPARγ in subcutaneous adipose tissue and (**B**) SFRP5, (**D**) WNT5A and (**F**) PPARγ in visceral adipose tissue of women with MO classified depending on NAFLD presence into NL or NAFLD. The data were normalized by NL group and expressed as mean standard error of the mean (SEM). SAT, subcutaneous adipose tissue; VAT, visceral adipose tissue; SFRP5, secreted frizzled-related protein 5; WNT5A, WNT family member 5a; PPARγ, peroxisome proliferator-activated receptor gamma; NW, normal weight; MO, morbid obesity; A.U arbitrary units. *p* < 0.05 was considered statistically significant (bold).

**Figure 5 ijms-23-09871-f005:**
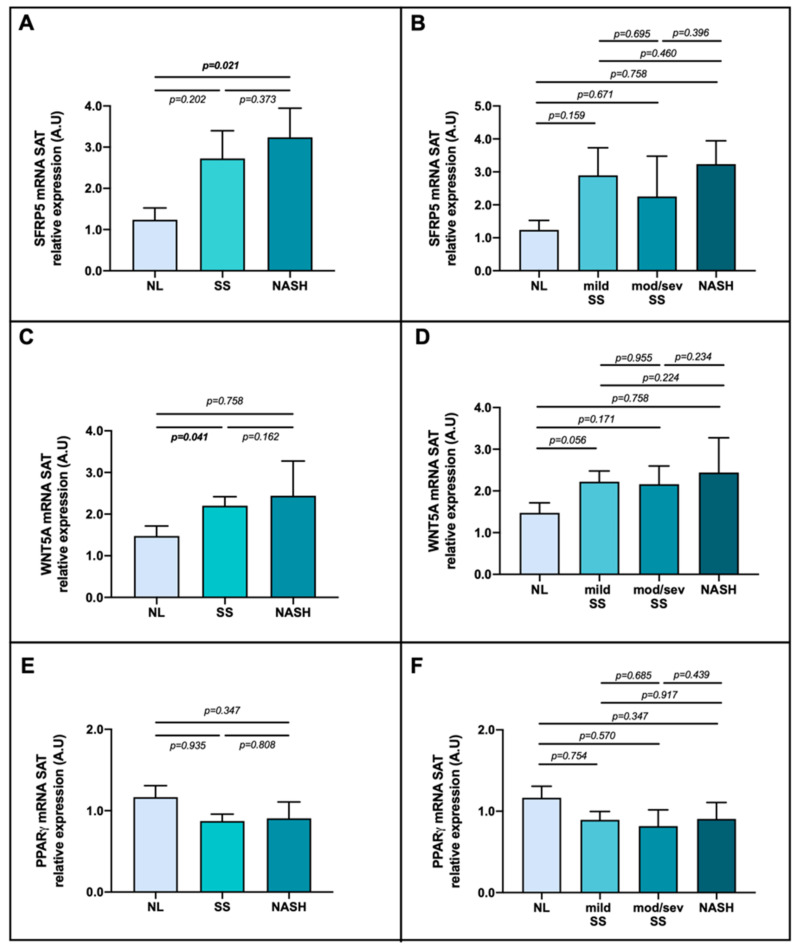
Differential relative mRNA abundance of (**A**) SFRP5, (**C**) WNT5A and (**E**) PPARγ in subcutaneous adipose tissue of women with MO classified by their liver histology as NL, SS and NASH; also, differential relative mRNA abundance of (**B**) SFRP5, (**D**) WNT5A and (**F**) PPARγ classified as NL, mild SS, mod/sev SS, and NASH depending on the histopathological groups. The data were normalized by NL group and expressed as mean standard error of the mean (SEM). SAT, subcutaneous adipose tissue; SFRP5, secreted frizzled-related protein 5; WNT5A, WNT family member 5a; PPARγ, peroxisome proliferator-activated receptor gamma; MO, morbid obesity; SS, simple steatosis; mod/sev SS, moderate and severe SS; NASH, nonalcoholic steatohepatitis; A.U arbitrary units. *p* < 0.05 was considered statistically significant (bold).

**Figure 6 ijms-23-09871-f006:**
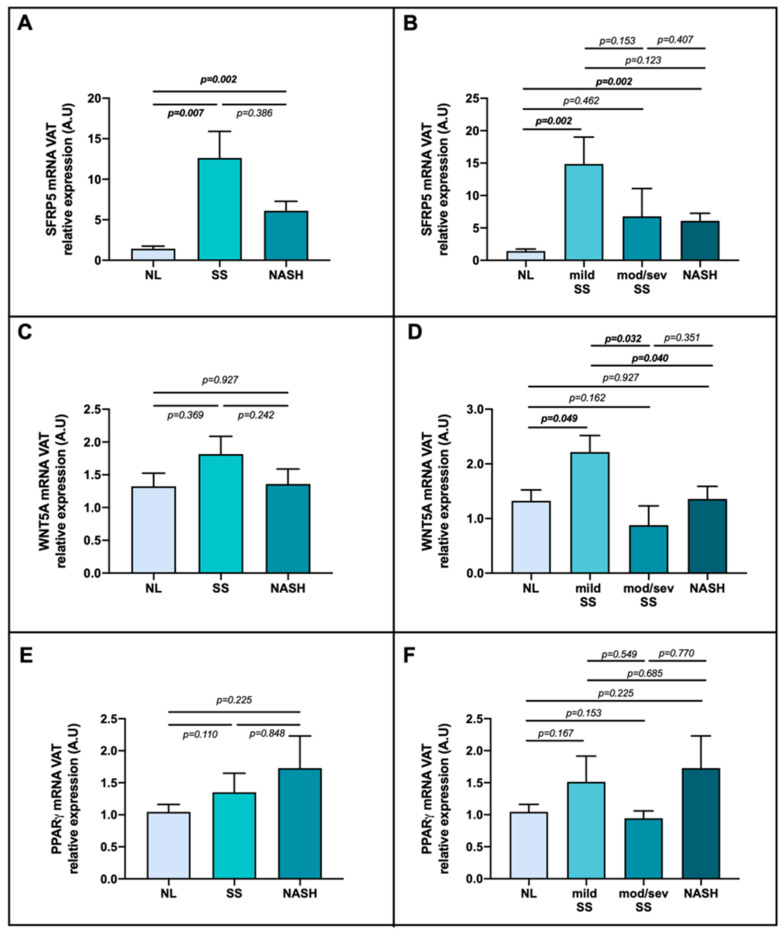
Differential relative mRNA abundance of (**A**) SFRP5, (**C**) WNT5A and (**E**) PPARγ in visceral adipose tissue of women with MO classified by their liver histology as NL, SS and NASH; also, differential relative mRNA abundance of (**B**) SFRP5, (**D**) WNT5A and (**F**) PPARγ classified as NL, mild SS, mod/sev SS, and NASH depending on the histopathological groups. The data were normalized by NL group and expressed as mean standard error of the mean (SEM). VAT, visceral adipose tissue; SFRP5, secreted frizzled-related protein 5; WNT5A, WNT family member 5a; PPARγ, peroxisome proliferator-activated receptor gamma; MO, morbid obesity; SS, simple steatosis; mod/sev SS, moderate and severe SS; NASH, nonalcoholic steatohepatitis; A.U arbitrary units. *p* < 0.05 was considered statistically significant (bold).

**Table 1 ijms-23-09871-t001:** Anthropometric and biochemical variables of women in the studied cohort.

	NW (*n* = 15)	MO (*n* = 60)		
Variables		NL (*n* = 20)	SS (*n* = 21)	NASH (*n* = 19)
Weight (kg)	57 (52–62)	119 (108–134) *	115 (111–129) *	110 (104–121) *
BMI (kg/m^2^)	22.47 (21.59–24.19)	43.62 (41.56–48.83) *	44.94 (42.07–46.85) *	44.54 (40.95–47.29) *
SBP (mmHg)	117 (110–125)	120 (100–132)	116 (108–127)	117 (105–132)
DBP (mmHg)	70 (66–75)	63 (58–71)	62 (59–73)	68 (60–78)
HOMA1-IR	1.30 (0.85–2.17)	2.07 (1.22–3.45)	2.44 (1.27–3.05) *	1.90 (1.45–6.16) *
Glucose (mg/dL)	83.50 (64.75–94.25)	85.50 (77.50–92.75)	93.00 (88.00–106.00) *^,$^	99.00 (82.25–109.75) *^,$^
Insulin (mUI/L)	7.00 (4.90–9.62)	9.70 (5.59–16.21)	9.80 (6.94–14.10) *	9.54 (5.68–26.02) *
HbA1c (%)	5.10 (4.70–5.40)	5.60 (5.30–5.75) *	5.55 (5.30–5.85) *	5.80 (5.28–6.43) *
TG (mg/dL)	86.00 (57.25–110.25)	106.00 (93.00–136.00) *	117.50 (84.00–165.50) *	130.50 (99.25–187.50) *
Cholesterol (mg/dL)	183.25 (163.53–209.50)	170.00 (150.15–214.50)	165.70 (132.75–189.50) *	162.00 (150.50–213.25)
HDL-C (mg/dL)	62.40 (48.45–73.00)	40.20 (31.50–48.50) *	43.50 (33.25–46.75) *	38.00 (34.50–44.00) *
LDL-C (mg/dL)	112.20 (89.90–130.00)	108.80 (95.20–141.80)	103.10 (70.20–124.85)	93.40 (79.30–126.83)
AST (UI/L)	20.00 (16.00–26.00)	19.50 (15.00–36.25)	21.00 (17.00–31.00) *	30.00 (18.00–43.50) *
ALT (UI/L)	17.00 (12.50–25.00)	21.00 (16.00–37.00)	29.50 (22.00–35.00) *	33.50 (18.75–41.00) *
GGT (UI/L)	14.00 (10.00–31.00)	17.00 (13.00–23.00)	21.00 (16.25–32.75)	26.00 (19.75–34.00) *^,$^
ALP (UI/L)	65.00 (51.50–88.00)	57.50 (47.75–71.75)	73.50 (62.00–86.00) ^$^	61.00 (53.25–74.50) ^a^

NW, normal weight; MO, morbid obesity; NL, normal liver; SS, simple steatosis; NASH, nonalcoholic steatohepatitis; BMI, body mass index; SBP, systolic blood pressure; DBP, diastolic blood pressure; HOMA1-IR, homeostatic model assessment method-insulin resistance; HbA1c, glycosylated hemoglobin; TG, triglycerides; HDL-C, high density lipoprotein cholesterol; LDL-C, low density lipoprotein cholesterol; AST, aspartate aminotransferase; ALT, alanine aminotransferase; GGT, gamma-glutamyltransferase; ALP, alkaline phosphatase. Data is expressed as the median (interquartile range). The Mann-Whitney test was used to find significant differences vs NW group (*), significant differences vs NL group (^$^) and significant differences vs SS group (^a^). *p* < 0.05 was considered statistically significant.

**Table 2 ijms-23-09871-t002:** Correlations between circulating cytokines and the relative mRNA expression of SFRP5, WNT5A or PPARγ in SAT and VAT of MO patients.

Variables	SAT			VAT		
	SFRP5	WNT5A	PPARγ	SFRP5	WNT5A	PPARγ
IL-1β (pg/mL)	ns	0.370 *	ns	0.404 *	ns	ns
IL-8 (pg/mL)	0.413 *	ns	ns	ns	ns	ns
IL-13 (pg/mL)	ns	ns	ns	0.445 **	0.345 *	ns
IL-17 (pg/mL)	0.435 *	ns	ns	0.362 *	ns	ns
IL-22 (pg/mL)	0.366 *	ns	ns	ns	ns	ns
Resistin (ng/mL)	−0.413 *	ns	ns	ns	ns	ns
Adiponectin (ng/mL)	−0.346 *	0.375 *	ns	ns	ns	ns
PAI (ng/mL)	ns	ns	ns	0.342 *	ns	ns
MCP-1 (pg/mL)	ns	ns	ns	ns	−0.377 *	ns

SAT, subcutaneous adipose tissue; VAT, visceral adipose tissue; SFRP5, secreted frizzled-related protein 5; WNT5A, WNT family member 5a; PPAR, peroxisome proliferator-activated receptor; R.E., relative expression; IL, interleukin; PAI, plasminogen activator inhibitor-1; MCP-1, monocyte chemoattractant protein-1; ns, non-significant correlation. Data are expressed as the correlation coefficient rho of Spearman. *p* < 0.05 was considered statistically significant (* *p* < 0.05; ** *p* < 0.01).

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
