# Peer review of "Increased Secreted Frizzled-Related Protein 5 mRNA Expression in the Adipose Tissue of Women with Nonalcoholic Fatty Liver Disease Associated with Obesity"

_ijms, 2022, doi:10.3390/ijms23179871_

Round 1

Reviewer 1 Report

In this present study, the authors have demonstrated differential mRNA levels of key  adipocytokines, i.e. Secreted frizzled-related protein 5 (SFRP5) and WNT family member 5A (WNT5A) in healthy and obese women subjects as far as NAFLD is concerned. In general, the study has been performed well and delivers some key observations (As opposed to the previous literature and findings) that add to the present knowledge of the SFRP5/WNT5A linked NAFLD progression, however, the study is not very conclusive. The manuscript mostly focuses on the results and lacks explanations for those observations. Here are some minor and major comments/concerns the authors need to address: First, the title of the manuscript needs to be more specific, not so broad. Also the abstract mostly describes the results and there has to be a conclusion/ key message at the end of the abstract. Second, in the case of obesity-related inflammation, WNT5A is upregulated and SFRP5 is down regulated in general (Fig 1). The authors have demonstrated SFRP5 mRNA levels are higher in NAFLD Vs NL subjects. That brings many questions; what triggers the abundance of SFRP5 in NAFLD subjects and what function SFRP5 is performing in VAT. No experiments have been done to address these questions. Third, the authors have mentioned that the very low-calorie diet that the subjects with MO followed three weeks prior to bariatric surgery might be a cause for the elevated SFRP5 levels. No experimental evidence has been demonstrated. The hypothesis needs to be validated in cell culture or animal model system. Lastly, the levels of all the key players of the pathway have been measured as mRNA levels, however, for further validation protein level data is necessary.

Reviewer 2 Report

This is an interesting study on the role of SFRP5 and WNT5A in NAFLD. It is a well designed study which is presented well and in a lot of detail.

Major comment:

It needs to be reinforced that the results shown here are associations (as acknowledged by the authors) and do not necessarily prove causation.

Minor comments:

1. All the study participants were women

2. How many patients were on the strict low calorie diet?

3. The figures can be improved because they currently show the statistically significant p values only. The non-significant p values should be added to the graphs.

4. it appears that subjects were allowed to drink up to 10 g of alcohol per day which may also affect results. I think this should be acknowledged in the limitations.

Reviewer 3 Report

In this study the authors evaluated the role of the SFRP5, WNT5A and PPARγ pathways in SAT and VAT tissues and their link with NAFLD pathogenesis in a cohort of NW and MO women with different levels of NAFLD related to obesity.

Based on the obtained results the authors suggested an association between adipose tissue SFRP5 expression and NAFLD related to obesity. In the study it was reported that the increased expression of SFRP5 in VAT seems to play a protective role in the first steps of NAFLD, while on other hand WNT5A is enhanced promoting liver damage. In conclusion the authors suggested that SFRP5 and WNT5A could play an important competitive role in NAFLD related to obesity and they could be therapeutic targets to prevent liver damage but mentioning that study needs further investigations.

This work is relevant to the field because the authors conducted the clinical study to examine the role of the SFRP5, WNT5A and PPARγ in NAFLD pathogenesis and to investigate the potential mode of action. The study is important because it is the first clinical study in this area and gave more data to this issue.

The study was well planned and done and the obtained data is presented in a good and clear manner.

The drawn conclusions are supported by the data presented in the paper.

The references are relevant and sufficient.

The manuscript is well written and easy to read but needs minor corrections.

In the Introduction section (lines 30) it should be added more information and explain in a couple sentences what are WNT and WNT5A and what is their role.

In the Materials and Methods section (line 424), it should be given information about the centrifuge used (company, city, state) and the same on line 431 for Bioplex 200 instrument (company, city, state).
